# A Corpus-Based Evaluation of Beamforming Techniques and Phase-Based Frequency Masking

**DOI:** 10.3390/s21155005

**Published:** 2021-07-23

**Authors:** Caleb Rascon

**Affiliations:** Instituto de Investigaciones en Matematicas Aplicadas y en Sistemas, Universidad Nacional Autonoma de Mexico, Circuito Escolar S/N, Mexico City 04510, Mexico; caleb.rascon@iimas.unam.mx

**Keywords:** evaluation, beamforming, AIRA, frequency masking, phase

## Abstract

Beamforming is a type of audio array processing techniques used for interference reduction, sound source localization, and as pre-processing stage for audio event classification and speaker identification. The auditory scene analysis community can benefit from a systemic evaluation and comparison between different beamforming techniques. In this paper, five popular beamforming techniques are evaluated in two different acoustic environments, while varying the number of microphones, the number of interferences, and the direction-of-arrival error, by using the Acoustic Interactions for Robot Audition (AIRA) corpus and a common software framework. Additionally, a highly efficient phase-based frequency masking beamformer is also evaluated, which is shown to outperform all five techniques. Both the evaluation corpus and the beamforming implementations are freely available and provided for experiment repeatability and transparency. Raw results are also provided as a complement to this work to the reader, to facilitate an informed decision of which technique to use. Finally, the insights and tendencies observed from the evaluation results are presented.

## 1. Introduction

Beamforming techniques have been popularly used to isolate a sound Source Of Interest (SOI) from the rest of the interferences in the auditory scene, only requiring to know a priori its Direction-Of-Arrival (DOA) in relation to the microphone array. They can also be used for sound source localization [1,2], by employing different candidate DOAs to locate the SOI. Considering the data flow of a typical auditory scene analysis application (where a sound source is located, then separated, and finally classified), beamforming techniques can be used to carry out the first two steps of that data flow in parallel. Additionally, they are usually implemented to be able to provide results in an online manner, meaning, that the amount of time it takes them to analyze a time window that has a length of *L* samples, is less than the time it takes to capture the next *L*-length time window. All of this in conjunction makes them very attractive for applications that involve online audio processing, such as robot audition [3,4] and auditory scene analysis [5].

Thus, it will be of benefit to the community to evaluate some of the more popular beamforming techniques in a systemic fashion, so as to provide a comparison between them and make the choice of which to use much more informed. To this effect, in this work, the AIRA corpus [6] is used to evaluate five popular beamforming techniques: Delay-and-Sum (DAS), Minimum Variance Distortion-less Response (MVDR), Linearly-Constrained Minimum Variance (LCMV), Generalized Sidelobe Canceller (GSC), and Geometric Source Separation (GSS). These were chosen because they are five of the most highly used techniques in the literature, all of which have been shown to provide acceptable separation performances, although (as far as the author knows) they have not been compared to each other in a repeatable, systemic fashion.

Alongside these techniques, a highly efficient beamforming-like technique is also evaluated, referred here as Phase-based Binary Masking (PBM). It employs a similar DOA-based multi-microphone separation paradigm, which is considerably less complex than most of the aforementioned beamforming techniques, and has provided good performances in simulated environments on its own [7]. It was chosen to provide a floor of comparison between complexity and performance.

The techniques were implemented under the same software libraries, evaluated using the same AIRA recordings, and their performance was measured using the same BSS Eval package [8]. The raw results are presented as a Appendix A to this work, which the reader is encouraged to use to inform their decision of which technique to employ in their audio processing applications. Additionally, the implementations of the techniques are freely-available at https://github.com/balkce/beamform (accessed on 22 June 2021) to facilitate the replication of the results presented in this work.

The contribution of this paper, as its purpose, is to present the tendencies observed during the corpus-based evaluation that the author considered important, as to:Reinforce well-known but unwritten facts of the aforementioned techniques.Provide new insights that should be considered by the reader when selecting a beamforming technique.

The paper is organized as follows. Section 2 describes all the evaluated techniques and their implementation, the evaluation methodology and resources, and the variables that were controlled during the evaluation. Section 3 presents the results accompanied by localized discussion. Section 4 discusses the results in an overall manner while providing some insights. Section 5 provides concluding remarks.

## 2. Materials and Methods

### 2.1. Evaluated Techniques

In this section, the evaluated techniques are explained. To this effect, it is assumed that there are a set of *M* input signals (x) which are commonly treated in the frequency domain (X). As briefly mentioned above, the aim of these techniques is to separate the Source Of Interest (SOI) from the rest of the acoustic scene, which is also commonly treated in the frequency domain (S). Essentially, the direction of arrival of the SOI (θ) is assumed to be known a priori.

Additionally, the position of each *m* microphone is known, where microphone m=1 is usually referred to as the “reference microphone”. To this effect, the Time Difference Of Arrival (TDOA) ($t_{m}$) is defined as the difference in time of arrival of S from the reference microphone to microphone *m*. A free- and far-field propagation is assumed, which results in a planar-wave of sound that simplifies the calculation of each $t_{m}$. Equation (Equation 1) presents the model that is usually followed [9]:(1)$$t_{m}=-\frac{d_{m}}{v_{sound}}cos\left(\theta_{m}-\theta \right)$$
where $v_{sound}$ is the speed of sound (∼343 m/s), $d_{m}$ is the distance of microphone *m* (in meters) to the reference microphone, and $\theta_{m}$ is the angle between the imaginary line intersecting the reference microphone and microphone *m* and the vertical axis of the frame of reference, as shown in Figure 1.

Having calculated each $t_{m}$, a steering vector ($W_{f}$) can be calculated for each frequency bin (*f*) as shown in Equation (Equation 2) [10].
(2)$$W_{f}=\left[\begin{array}{c} 1\\ e^{-i2\pi ft_{2}}\\ e^{-i2\pi ft_{3}}\\ \ldots \\ e^{-i2\pi ft_{M}} \end{array}\right]$$

For simplicity purposes, each steering vector is “binned” at the microphone level, such that $W_{f}\left[m\right]=e^{-i2\pi ft_{m}}$. In addition, since the TDOA of the reference microphone to itself ($t_{1}$) is 0, $W_{f}\left[1\right]$ is always 1. Because the TDOA is essential to the inner workings of beamforming techniques, the minimum number of microphones be used is 2.

The steering vector models the effect of the TDOA of the SOI of each microphone at each frequency bin [10], such that:(3)$$X_{m}\left[f\right]=S\left[f\right]W_{f}\left[m\right]$$

In addition, for simplicity, Equation (Equation 4) defines the term X[f] as the *f* column of the input matrix
(4)$$X\left[f\right]=\left[\begin{array}{c} X_{1}\left[f\right]\\ X_{2}\left[f\right]\\ X_{3}\left[f\right]\\ \ldots \\ X_{M}\left[f\right] \end{array}\right]$$

Having established the general terminology and models, the explanation of each technique is as follows.

#### 2.1.1. Delay-and-Sum (DAS)

The Delay-and-Sum (DAS) beamformer [11,12] is the simplest technique evaluated here. It aims to first “align” the input signals to compensate for the TDOAs at each microphone and add (although an averaging is usually carried out) the resulting “aligned” signals so as to obtain a type of “constructive interference” at its output. Thus, the energy of every source in the acoustic scene that is not in the path of θ will be diminished by the number of microphones *M*. This can be simply carried out by applying a complex conjugate to each steering vector at each frequency bin, and then carrying out a vector multiplication with the frequency bins of all the input signals, as shown in Equation (Equation 5).
(5)$$Y_{DAS}\left[f\right]=\frac{W_{f}^{H}X\left[f\right]}{M}$$
where the $.^{H}$ operator is the complex conjugate, such that: ${\left[\begin{array}{c} a+ib\\ c-id \end{array}\right]}^{H}=\;\left[a-ib,c+id\right]$. Broadly speaking, this operator is compensating for the TDOA effects at each microphone by shifting each input signal in exactly the opposite manner as each $t_{m}$.

#### 2.1.2. Minimum Variance Distortion-Less Response (MVDR)

The Minimum Variance Distortion-less Response (MVDR) beamformer [13,14,15] aims to obtain a set of optimized weights (A) that, in conjunction minimize the energy of Y, except in the direction of θ.

In this way, they can be applied directly to the input signals, similar to DAS, as shown in Equation (Equation 6).
(6)$$Y_{MVDR}\left[f\right]=A_{f}^{H}X\left[f\right]$$

These weights are obtained by establishing the energy output as a function of the covariance of the input signals, as shown in Equation (Equation 7).
(7)$$Y_{MVDR}{\left[f\right]}^{2}={\left(A_{f}^{H}X\left[f\right]\right)}^{2}=\left(A_{f}^{H}X\left[f\right]\right)\left(X{\left[f\right]}^{H}A_{f}\right)=A_{f}^{H}R_{f}A_{f}$$
where $R_{f}$ is the covariance of the input signals at frequency *f*. The last term in Equation (Equation 7) is minimized, while maintaining the restriction presented in Equation (Equation 8).
(8)$$W_{f}^{H}A_{f}=1$$

This results in Equation (Equation 9).
(9)$$A_{f}=\frac{R_{f}^{-1}W_{f}}{W_{f}^{H}R_{f}^{-1}W_{f}}$$

An important implementation consideration for MVDR is that the estimation of $R_{f}$ should be robust so as to produce good separation results. This requires the use of past time windows for each frequency *f*, to calculate ${\hat{R}}_{f}$ as shown in Equation (Equation 10)
(10)$${\hat{R}}_{f}=\left[X{\left[f\right]}_{0},X{\left[f\right]}_{1},X{\left[f\right]}_{2},\ldots ,X{\left[f\right]}_{P-1}\right]{\left[X{\left[f\right]}_{0},X{\left[f\right]}_{1},X{\left[f\right]}_{2},\ldots ,X{\left[f\right]}_{P-1}\right]}^{H}$$
where $X{\left[f\right]}_{p}$ is the X[f] of the *p*th time window. As can be deduced, using more windows (*P*) to calculate ${\hat{R}}_{f}$ results in a more robust result, but it will take more time to calculate. This, in conjunction with the need to also reverse this matrix, results in MVDR requiring a considerable of computing power to be run in an online manner.

#### 2.1.3. Linearly-Constrained Minimum Variance (LCMV)

Linearly-Constrained Minimum Variance (LCMV) beamformer [16,17] is an evolution of MVDR that also considers the presence and locations of the interferences that are present in the acoustic scene. Thus, the optimized weights (A) that are to be calculated aim to:Minimize the energy of Y, except in the direction of θ.Cancel the energy in the direction of known interferences ($\theta_{I}$).

To this effect, a similar minimization paradigm to MVDR’s is used, but now with the restriction shown in Equation (Equation 11).
(11)$${\left[W_{f},W_{I}f\right]}^{H}A_{f}=\left[1,0,0,\ldots \right]$$
where $W_{I}f$ is a set of steering column vectors, each calculated from a value of $\theta_{I}$. The optimization results in an equation similar to Equation (Equation 9), as shown in Equation (Equation 12).
(12)$$A_{I}f=\frac{R_{f}^{-1}W_{f}}{W_{f}^{H}R_{f}^{-1}W_{f}}$$
where $A_{I}f=\left[A_{f},A_{I1}f,A_{I2}f,\ldots \right]$, i.e., the first column of $A_{I}f$ bares the optimized weights to be applied to X[f].

#### 2.1.4. Generalized Sidelobe Canceller (GSC)

Generalized Sidelobe Canceller (GSC) beamformer [18,19] is a generalization of LCMV [20] and aims to cancel the energy of all directions except that of θ. To do this, it follows the diagram shown in Figure 2.

Although it can be carried out in the frequency domain, it is detailed here in the time domain for ease of explanation. Having established this, it is important to note that the upper DAS component can be carried out in the frequency domain, since it is independent from the lower noise estimation component.

The lower component is arguably the main factor of GSC. As can be seen, it uses a blocking matrix, the outputs of which are defined in Equation (Equation 13).
(13)$$B_{m}\left[\tau \right]=\left[x_{d:m+1}\left[\tau \right]-x_{d:m}\left[\tau \right]\right]$$
where $B_{m}$ is the *m*th output of B and $x_{d:m}$ is the delayed *m* input signal in a manner similar to what is carried out in DAS, but without the addition or averaging part of the process. It can be observed that B has M−1 outputs. It is important to note that this blocking matrix is one of many that can be used, but it provides a good balance between simplicity and carrying out well the objective of an “anti-beamformer”.

These outputs are then fed to a set of filters the sum of the outputs of which aim to estimate the accumulative interference in the acoustic scene such that it is able to be canceled from the DAS output by a simple subtraction. Since the attributes of an environment, as well as the acoustic characteristics of the sound sources in the scene, are usually non-stationary, these filters are estimated in an online manner similar to the Least Means Squares (LMS) method. This method follows the steepest gradient from both the generated output and the estimated noise, as shown in Equation (Equation 14).
(14)$$G_{m}\leftarrow G_{m}+\mu \left[\begin{array}{c} y_{GSC}\left[\tau \right]x_{d:m}\left[\tau \right]\\ y_{GSC}\left[\tau -1\right]x_{d:m}\left[\tau -1\right]\\ \ldots \\ y_{GSC}\left[\tau -K+1\right]x_{d:m}\left[\tau -K+1\right] \end{array}\right]$$
where τ is the time index and *K* is the length of $G_{m}$. As can be deduced, to update the filters, $y_{GSC}\left[\tau \right]$ first needs to be calculated by following Equation (Equation 15).
(15)$$y_{GSC}\left[\tau \right]=\sum_{m=1}^{M}x_{d:m}\left[\tau \right]-\sum_{m=1}^{M-1}\sum_{k=0}^{K-1}G_{m}\left[k\right]x_{d:m}\left[\tau -k\right]$$

To provide stability in the convergence of the LMS algorithm, the value of μ can vary depending on the energy of the beamformer output and of the estimated noise [21] (pg. 42–43). Thus, instead of using μ, the value $\mu_{m}$ is calculated for each filter $G_{m}$ at each τ, as shown in Equation (Equation 16).
(16)$$\begin{array}{c} {\left|y_{GSC}\left[\tau :K\right]\right|}^{2}=\frac{\sum_{k=0}^{K-1}y_{GSC}{\left[\tau -k\right]}^{2}}{K}\\ {\left|x_{d:m}\left[\tau :K\right]\right|}^{2}=\frac{\sum_{k=0}^{K-1}x_{d:m}{\left[\tau -k\right]}^{2}}{K}\\ \mu_{m}=\left\{\begin{array}{cc} \frac{\mu_{0}}{{\left|y_{GSC}\left[\tau :K\right]\right|}^{2}}, & \mathrm{if}\;\mu_{0}\frac{{\left|x_{d:m}\left[\tau :K\right]\right|}^{2}}{{\left|y_{GSC}\left[\tau :K\right]\right|}^{2}}<\mu_{max}\\ \frac{\mu_{0}}{{\left|x_{d:m}\left[\tau :K\right]\right|}^{2}}, & \mathrm{otherwise} \end{array}\right. \end{array}$$
where $\mu_{0}$ is a type of “starting point” for $\mu_{m}$ and $\mu_{max}$ is the maximum value of $\mu_{m}$. Usually, $\mu_{max}$ is several orders of magnitude greater than $\mu_{0}$.

#### 2.1.5. Geometric Source Separation (GSS)

Geometric Source Separation (GSS) is a type of hybrid approach to beamforming [22], since it also involves a blind source separation paradigm. To this effect, this technique aims to separate all of the sound sources in the acoustic scene, of which it assumes to know their DOAs (similar to LCMV). It is important to note that the variation of the technique described here is the one that was proposed to be run in an online manner [3].

GSS aims to find a set of weights ($A_{f}$) for each separated source that satisfies the following restrictions:The separated sources are minimally correlated to each other, similar to Principal Component Analysis (PCA) [23,24,25].$W_{f}^{H}A_{f}=1$ for each separated source.

Because of the blind source separation paradigm, the set of $A_{f}$s for all separated sources ($\alpha_{f}$) can be considered as the de-mixing matrix for frequency *f*. To this effect, $\Omega_{f}$ is matrix that can be considered as a type of “multiple steering vector”, since each of its columns is the $W_{f}$ of each separated source.

To estimate $\alpha_{f}$, GSS employs an LMS adaptation paradigm (similar to GSC), with two gradients to minimize ($J1_{f}$ and $J2_{f}$), as shown in Equation (Equation 17).
(17)$$\begin{array}{cc} Y_{GSS}\left[f\right] & =\alpha_{f}X\left[f\right]\\ R_{f:Y} & =Y_{GSS}\left[f\right]Y_{GSS}{\left[f\right]}^{H}\\ E_{f} & =R_{f:Y}-diag\left(R_{f:Y}\right)\\ J1_{f} & =4\left(E_{f}\alpha_{f}X\left[f\right]\right)X{\left[f\right]}^{H}\\ J2_{f} & =2\left(\alpha_{f}\Omega_{f}-I\right)\Omega_{f}^{H}\\ \alpha_{f} & \leftarrow \alpha_{f}-\mu \left(\sqrt{{\left||X\left[f\right]|\right|}^{2}}J1_{f}+J2_{f}\right) \end{array}$$
where I is the identity matrix and diag is an operator that extracts the elements from the diagonal of a matrix and sets all other elements to zero. As can be deduced, minimizing $J1_{f}$ achieves the first aforementioned restriction (low correlation), and minimizing $J2_{f}$ achieves the second (parallel with steering vectors).

It is important to mention that to make the output of GSS comparable to the other beamforming techniques, only the first sound source is returned as the SOI. Because the order in which the sources appear in $\Omega_{f}$ is the same order in which they appear in $Y_{GSS}\left[f\right]$, this amendment should not hinder its performance, while making it appropriate to evaluate GSS alongside the rest of the techniques.

#### 2.1.6. Phase-Based Binary Masking (PBM)

Although not technically a beamforming technique, Phase-based Binary Masking (PBM) [7] employs a paradigm similar to beamforming to create binary mask that aims to extract the SOI from the reference microphone. The reasoning behind also evaluating this technique is to provide a type of complexity floor (other than DAS) that may perform comparably to other much more complex techniques.

The input signals are first aligned using the steering vector $W_{f}$, but without adding or averaging them together. This results in a set of $X_{d:m}$, one for each microphone (except, of course, the reference microphone). Then, for each frequency bin *f*, an average phase difference between all possible microphone pairs is calculated ($\delta_{f}$), as shown in Algorithm 1.
**Algorithm 1:** Calculated the average phase difference in frequency *f*.
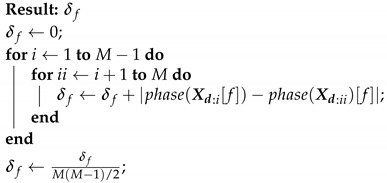


The binary mask (β) is then created by establishing an upper phase difference threshold (σ) while applying Equation (Equation 18).
(18)$$\beta \left[f\right]=\left\{\begin{array}{cc} 1, & \mathrm{if}\;\delta_{f}<\sigma \\ 0, & \mathrm{otherwise} \end{array}\right.$$

That is to say, if $\delta_{f}$ is below σ, it means that the frequency component *f* is aligned throughout all the microphones towards θ, which implies that it belongs to the SOI. Otherwise, it means that the frequency component *f* is not aligned towards θ, which implies that it belongs to an interference.

The output is then calculated as shown in Equation (Equation 19).
(19)$$Y_{PBM}\left[f\right]=X_{1}\left[f\right]\beta \left[f\right]$$

The reader may find it interesting that the binary nature of this technique can also provide the estimation of the accumulative interference in the acoustic scene. This is achieved by reverting the conditional values in Equation (Equation 18). Given the efficiency of the technique, both estimations (of the SOI and the accumulative interference) can be provided simultaneously.

It is important to note that this technique is widely used in the acoustic sector (similar to how DAS is used in the audio processing sector, as well as in the antenna array sector). Furthermore, similar to DAS, it has been extended and enhanced in different manners. To this effect, the binary nature of this technique results in a “hard” frequency mask that can sometimes provide some unwanted “musical” artifacts. This mask can be softened by assuming a frequency-dependent phase variance and empirically accounting for it, as shown in [26]. Another strategy is to employ an interference-leakage removal strategy that requires the estimation of the frequency co-variance matrix, as shown in [27]. However, both of these variants seem to be much more complex than just using the binary mask and are even comparable in complexity to the evaluated MVDR, LCMV, and GSS. Evaluating these variants would defeat the purpose of evaluating a much simpler technique (other than DAS) that may offer comparable performance to other much more complex techniques. Thus, the technique that only employs Algorithm 1 and Equations (Equation 18) and (Equation 19) is evaluated.

### 2.2. Implementation

To facilitate the comparison between the aforementioned techniques during their evaluation when filtering audio signals in an online manner, as well as the reproducibility of the evaluation results, all techniques were implemented using the following software libraries:JACK Audio Connection Kit (JACK) [28]: It is an audio server that can provide connectivity between audio agents, while providing low latency. This library provides direct access, with near real-time response, to multi-channel synchronous input and output audio signals. In addition, it also provides a transparent manner to switch between evaluation mode (audio files are fed to the beamformers; detailed later) and live mode (the beamformers are connected to live hardware).Robot Operating System (ROS) [29]: It is a framework that provides a structured communication layer between software modules. This library provides an easy-to-implement mechanism to launch the beamformers and communicate with them while they are running. It substantially simplified the automation of the evaluation.

Using both libraries required the implementation of a “bridge” between them, so that information could flow bi-directionally from one framework to another. This bridge, referred to as “rosjack”, is the basis of the implementation of all the techniques, which are freely-available at https://github.com/balkce/beamform (accessed on 22 June 2021).

### 2.3. Evaluation Methodology

The evaluation is based on the Acoustic Interactions for Robot Audition (AIRA) corpus [6] (detailed below), whereby a set of real-life multi-channel recordings is fed to the evaluated techniques in an online manner. Each recording captured several active sources being reproduced by electronic speakers, as to present the case of a multiple-source acoustic environment, typical in real-life settings.

To feed these recordings in an online manner, a JACK agent was developed to read the recording files, and feed a series of *L*-length time windows to the techniques; after each time window was fed, the JACK agent waited an amount of time equivalent to the capture of the next *L*-length time window to feed the next time window. The output was stored and, in conjunction with the pristine recordings included in AIRA, it was used to obtain a set of evaluation metrics (also detailed below) to measure its separation performance. In this section, an overview of the evaluation resources used for the evaluation is presented.

#### 2.3.1. Acoustic Interactions for Robot Audition (AIRA)

The Acoustic Interactions for Robot Audition (AIRA) corpus [6] is a set of real-life multi-channel 30 s recordings accompanied by the pristine recordings of the separated sources, which is aimed to be used as the ideal result to achieve when attempting separating the sources from the multi-channel recordings. Each channel is mapped to a specific microphone, the location of which (in relation to the rest of the microphones) is known. Additionally, the location of all active to-be-separated sources are also known. AIRA was recorded with different microphone array geometries, as well as in different acoustic environments, with varying amounts of active sound sources (up to 4).

The sound sources were positioned at 1 m from the center of the microphone array, with a varying amount of angular separation, ranging from 30${}^{\circ }$ to 120${}^{\circ }$. An example of how the sources were located around the microphone array is provided in Figure 3.

For this evaluation, two acoustic environments were used:**Anechoic Chamber**: This environment is located inside the full-anechoic chamber [30] of the *Instituto de Ciencias Aplicadas y Tecnologia* (ICAT, formerly known as the *Centro de Ciencias Aplicadas y Desarrollo Tecnologico*, CCADET) of the *Universidad Nacional Autonoma de Mexico* (UNAM). It measures 5.3 m × 3.7 m × 2.8 m. It has a very low noise level (≈0.13 dB SPL) with an average reverberation time <0.01 s. No other noise sources were present in this setting. Recordings have a SNR of ≈43 dB. The microphone pre-amplification was set at 0 dB, and the the speaker amplification was set at −30 dB to compensate for the maximum SPL difference between monitors.**Cafeteria**: This environment is a cafeteria located inside the UNAM campus and was used during a 5 h period of high customer presence. It has an approximate size of 20.7 m × 9.6 m × 3.63 m. It has a high noise level (71 dB SPL) with an average reverberation time of 0.27 s. Its ceilings and floor are made of concrete, and its walls are made of a mixture of concrete and glass. Noise sources around the array included: people talking, babies crying, tableware clanking, some furniture movement, and cooling fans of stoves. The recordings in this environment have a Signal-to-Noise Ratio (SNR) of ∼16 dB.

The channels used in the evaluation are the six first microphones of the three-dimensional array shown in Figure 4, since they do not violate the free- and far-field assumption in the propagation model described in Section 2.1.

The positions of the microphones used are presented in Table 1.

The audio interface used was the *Behringer X-32* [31]: a digital mixer that can act as an external audio interface, providing support for 32 XLR inputs and 32 XLR outputs. The microphones used were *Shure MX393/O* [32], which are omnidirectional with a flat frequency response across the vocal range and provide an SNR of 80 dB with a reference signal of 94 dB SPL.

All recordings are sampled at 48,000 Hz; thus, JACK was configured such that it ran with that sampling frequency. For more details of how the AIRA corpus was captured, its full documentation is available at https://aira.iimas.unam.mx/ (accessed on 22 June 2021).

As for the evaluation itself, the computer equipment used to run the actual evaluation bares an i7-4790 CPU which has 8 cores and a boost clock of 4.00 GHz, as well as 16 GB of RAM. It was chosen because of its considerable amount of computing power, which should provide little bottleneck to run beamforming techniques in an online manner.

#### 2.3.2. Evaluation Metrics

To measure the performance of the results obtained by the evaluated techniques, the Signal-to-Interference Ratio (SIR) was used and calculated using the BSS Eval toolkit [8], which defines it as shown in Equation (Equation 20).
(20)$$SIR=10log_{10}\frac{\left|\right|\hat{S}{\left|\right|}^{2}}{{\left||N|\right|}^{2}}$$
where $\hat{S}$ is the estimation of the source of interest *S* and *N* is the accumulative interferences in the environment. To calculate this last term, the pristine recordings are required, which are included in the AIRA corpus.

Along with the SIR metric, BSS Eval can also calculate the Signal-to-Artifact Ratio (SAR) and the Signal-to-Distortion Ratio (SDR). Although the results in this work (presented in Section 3) are centered on the SIR metric, both the SAR and the SDR were also registered in the extended CSV file included as part of this paper.

In addition to the SIR, it is also of interest to know if the technique is able to run in an online manner. As mentioned above, to run in an online manner, the technique should be able to deliver the results in a time frame shorter than the time it takes to capture the next time window (*L*). If a technique is not able to do so, an “overrun” is triggered for each time window it is not able to analyze in such a time frame. To this effect, JACK can report the number of overruns the technique triggered during each of its runs, which was registered as part of the evaluation.

#### 2.3.3. Evaluation Variables

**Acoustic Environments.** As mentioned above, two acoustic environments were used: a noise-less *Anechoic Chamber* and a noisy *Cafeteria*.**Number of Interferences.** Since AIRA has a maximum of 4 active sources, the SOI was considered one of those sources and the rest were interferences (ranging from *1* to *3*). It is important to mention that, with each set of recordings, the technique was evaluated the same number of times as the number of sources, with each evaluation considering each active source as the SOI and the rest as interferences.**Number of Microphones.** As mentioned above, the six first microphone of the three-dimensional array were used since they do not break the free-field assumption. Thus, the number of microphones ranged from *2* to *6*.**Direction-of-Arrival Error.** To evaluate the robustness of the techniques against an erroneous DOA of the SOI, an error was artificially introduced ranging from *0${}^{\circ }$* to *25${}^{\circ }$*.

#### 2.3.4. Technique Parameters during Evaluation

*L* is set at 1024 samples per time window. To carry out online audio analysis, the Short-Time Fourier Transform (STFT) was used, as well as the weighted overlap-and-add technique (WOLA). A 2∗L Hann window was applied before converting to the frequency domain, and the square root of a 2∗L Hann window was used as the synthesis window after converting back to the time domain. All delays were carried out using the $e^{-i2\pi ft}$ Fourier operator.

The following lists the parameters used for each evaluated technique:**DAS.** No parameters were required to be set.**MVDR and LCMV.** Ten windows were used to calculate R, and only frequencies between 100 and 16,000 Hz and that have a normalized amplitude above 0.001 were analyzed.**GSC.**$\mu_{0}$ was set at 0.001, $\mu_{max}$ was set at 0.1, and the filter length (*K*) was set at 128.**GSS.** One window was used to calculate R, only frequencies between 100 and 16,000 Hz and that have a normalized amplitude above 0.001 were analyzed, and μ was set at 0.001.**PBM.** Only frequencies that have a normalized amplitude above 0.001 were analyzed, and the upper phase difference threshold (σ) was set at 20${}^{\circ }$.

## 3. Results

### 3.1. Number of Interferences

As mentioned above, the number of sources in AIRA ranges 2–4, with one of them being the SOI and the rest considered as interferences. For simplicity, the results shown are presented with no DOA error, with the minimum and maximum number of microphones (to present both edge cases), as well as in both acoustic environments (the Anechoic Chamber and the Cafeteria) in Figure 5.

As can be seen, there is a clear tendency in all cases that increasing the number of interferences results in a decrease of separation performance. It also presents the first case, in which it can be observed that, in the Cafeteria environment (which is more noisy and with more reverberation), there is a considerably lower separation performance than in the Anechoic Chamber.

It can also be observed that increasing the number of microphones increases performances, which is explored in more detail in the following section.

### 3.2. Number of Microphones

As mentioned above, the number of microphones used in this evaluation ranges 2–6. For simplicity, the results shown are presented with no DOA error, with the minimum and maximum number of sources (to present both edge cases), as well as in both acoustic environments (the Anechoic Chamber and the Cafeteria) in Figure 6.

As can be seen, there is a clear tendency in all cases that increasing the number of microphones results in an increase of separation performance. It also presents the second case in which the Cafeteria presents a considerably lower separation performance than in the Anechoic Chamber.

It can also be observed that increasing the number of sources decreases performances, as explored in the previous section.

### 3.3. Direction-of-Arrival Error

To explore the impact of the DOA error in the separation performance, while maintaining an easily observed result, the results shown in Figure 7 are from the edge case that provides the better performances from all techniques: in the Anechoic Chamber, using six microphones and two sources.

As can be seen, there is a tendency that increasing the DOA error decreases the separation performances. However, some techniques may seem to not be as affected as others. It is possible that, because of their relative low overall performance, there is a type of “performance floor” that is being reached and, thus, the impact of the DOA error is not being observed. To this effect, an edge case that provides bad performances from all techniques is shown in Figure 8: the Cafeteria environment. The same numbers of microphones and sources are used as in Figure 7 for comparison sake.

As can be seen, the “performance floor” is also being reached, and, thus, the impact of the DOA error is not being shown. This phenomenon is also observed in the following section. However, the tendency that a high DOA error results in a low SIR performance can still be observed with high-performing techniques (mainly PBM).

### 3.4. Overall Results Variation

As can be concluded at this point, the overall separation performance of all techniques is relatively quite low (<10 dB) in comparison with state-of-the-art techniques which are able to achieve SIR results above 10 dB in real-life settings [33]. However, it is important to note that these results are obtained as the average of the SIR metric. To provide a wider view of the last results, in Figure 9, a boxplot is shown of the SIR variation. Each boxplot represents the four quartiles of the SIR results distribution: the bottom vertical line represents the bottom 25%; the box, the middle 50%; the top vertical line, the top 25%; each red cross, an outlier; the horizontal line, the average. A boxplot is shown for four different edge cases: in both Anechoic Chamber and Cafeteria acoustic environments, with the minimum and maximum number of microphones, two sources, and no DOA error (to simplify the visualization of the results).

As can be seen, there is a considerable amount of variation in the SIR results in all the techniques. However, it can also be seen that increasing the number of microphones reduces the variation of the results, even if the average SIR is not as affected. Additionally, the aforementioned “performance floor” is also observed: the variation is smaller in the Cafeteria than in the Anechoic environment, while the average SIR only drops a few dB. This means that in the Anechoic environment most techniques are performing well (>10 dB) in a considerable amount of cases, which is not true in the Cafeteria environment.

### 3.5. Number of Overruns

It is important to also explore the ability of these techniques to run in an online manner, by measuring the number of overruns that each technique triggered. To this effect, the average number of overruns triggered per run (alongside their standard deviation) separated by acoustic environment, for each technique, is shown in Table 2.

Since *L* is 1024 samples, and JACK is configured with a sampling frequency of 48,000 samples per second, one window is equivalent to 0.0213 s. Thus, 10 overruns amounts to 0.213 s, which is lower than 1% of a 30 s. recording, which can be considered acceptable. To this effect, as can be seen, although MVDR and LCMV provide more overruns than all the other techniques, the number of overruns they trigger in the Anechoic environment is still acceptable (<10). However, in the Cafeteria environment, the numbers of overruns are considerably higher (by several orders of magnitude). This can be explained by the manner in which the online implementations of MVDR and LCMV choose which frequencies to analyze, which involve not only the frequency value but also their energy. In a noisy environment, there are more high-energy frequencies to analyze than in a low-noise environment.

In addition, controlling for the number of overruns triggered (such that only runs with no overruns triggered are used to calculate the evaluation results) makes it so that LCMV and MVDR do not have any results to report. Thus, one of the reasons behind their low performances may be their high number of overruns triggered. It is important to note that this finding is consistent to what has been observed in the literature [34].

## 4. Discussion

Although there are considerably low overall performances throughout the evaluation, it is important to remember that these techniques were evaluated when running in an online manner. Many of the state-of-the-art source separation techniques (mainly based on deep learning) do not run in such a way, opting to be fed full audio recordings [33,35]. This lends to higher performances since information in the future of the current window can also be utilized to obtain a good separation performance. This is not the case with online techniques, since only information in the past of the current window can be utilized. In addition to this, it is important to note that the results shown in Section 3.5, where some techniques (namely, GSC, GSS and PBM) presented a considerable number of results near or even above the 10 dB mark in the Cafeteria environment with six microphones. This is a feat comparable with the state-of-the-art technique in offline mode, although many only using one microphone/channel [33,35].

It is also important to note that PBM consistently outperformed all other techniques in all circumstances, even though its implementation is considerably less complex. This makes it an attractive alternative to far more complex techniques such as MVDR, LCMV, and GSC. To this effect, even though, as shown in Figure 9, PBM does provide more variation in its results, it consistently provides a result distribution that is overall above the rest of the rest of the techniques. In fact, it can be seen that, in the case of the Anechoic environment with six microphones, the top 75% quartile is above the average (50%) of all other techniques. In the Cafeteria environment with six microphones, this lead diminishes in some cases, but its top 50% quartile is still above the 75% quartile of the rest, and in the other cases this lead was maintained (in the case of MVDR and LCMV). It is important to note, however, that, in both environments, when using only two microphones, this lead diminishes considerably, which implies that PBM is highly sensitive to the number of microphones, a fact that is explicitly shown in Figure 6.

It is also worth mentioning that GSS, which is a technique that is popularly used in audio processing frameworks such as ManyEars [2] and Hark [36], did not provide competitive results, even compared to the simpler PBM. It is important to note that GSS is usually paired with a multi-channel post-filter that, as the authors put it: “for a given channel output of the GSS, the transient components of the corrupting sources are assumed to be due to leakage from the other channels during the GSS process” [3]. This implies that the post-filter provides a type of refinement of the separation results, which aims to reduce the presence of the interferences for each GSS output, which in turn may increase the SIR performance of their combination. All of this in conjunction implies that GSS by itself may not be enough to obtain good SIR performance, and that the post-filter is a requirement. The evaluation of GSS with and without this post-filter is left for future work.

## 5. Conclusions

Beamforming techniques are compatible with many types of auditory scene analysis applications which require separating the source of interest from the acoustic scene in an online manner. Thus, a thorough evaluation and comparison between such techniques can benefit the community when making an informed decision of which technique to use.

The AIRA corpus was used to evaluate five popular beamforming techniques: Delay-and-Sum (DAS), Minimum Variance Distortion-less Response (MVDR), Linearly-Constrained Minimum Variance (LCMV), Generalized Sidelobe Canceller (GSC), and Geometric Source Separation (GSS). Along these techniques, a highly efficient beamforming-like technique was also evaluated, referred here as Phase-based Binary Masking (PBM). All evaluated techniques were implemented under the same software libraries (JACK, ROS, and an in-house communication bridge). These implementations are freely-available at https://github.com/balkce/beamform (accessed on 22 June 2021).

It was shown that, on average, all beamforming techniques performed poorly under a real-life Cafeteria-type environment, although PBM outperformed all of them under all circumstances. Additionally, it was observed that increasing the number of interferences reduced performance; increasing the number of microphones improved performance; increasing the error of the direction-of-arrival of the source of interest decreased performance (although most techniques reached a “performance floor” that made this tendency difficult to observe).

However, when observing the variation of the evaluation results, it was shown that for most techniques the distribution of their results had a considerable portion in ranges that were on par with the state-of-the-art method even when running in an online manner (which the state-of-the-art method mainly does not). Thus, it can be concluded that, with a moderate number of sources and microphones, beamforming techniques can still provide acceptable separation performance. As for which technique to use, the author found PBM an attractive choice, since it outperformed all the other techniques in all circumstances, while being far less complex (except for DAS, which is comparable in complexity). Having mentioned this, it is important to note that the raw results, which are the basis of all the figures and tables in this work, are available as a Appendix A, and readers are encouraged to use it to inform their choice of technique in their own audio processing implementations.

## Figures and Tables

**Figure 1 sensors-21-05005-f001:**
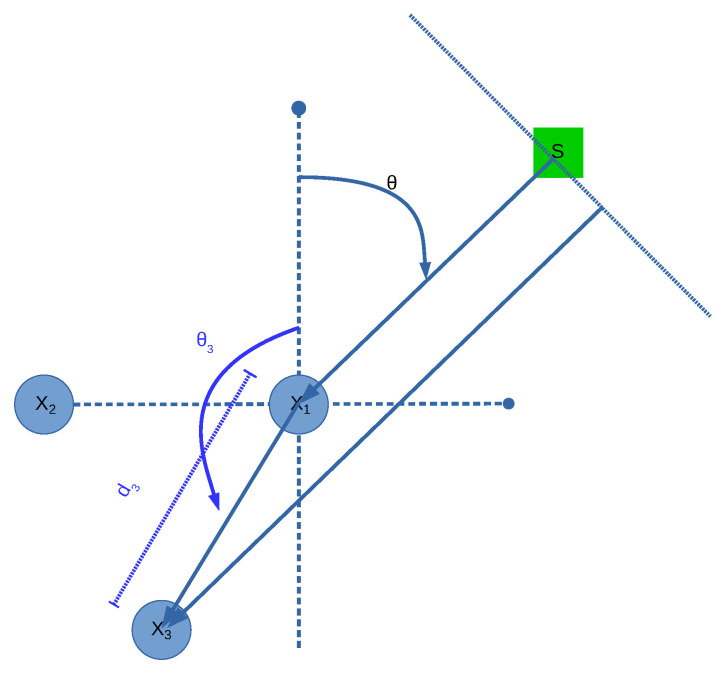
A possible frame of reference, in this case a triangular array, where $\theta_{3}$ is exemplified. The green square is the SOI (S) and each blue circle is a microphone ($X_{m}$).

**Figure 2 sensors-21-05005-f002:**
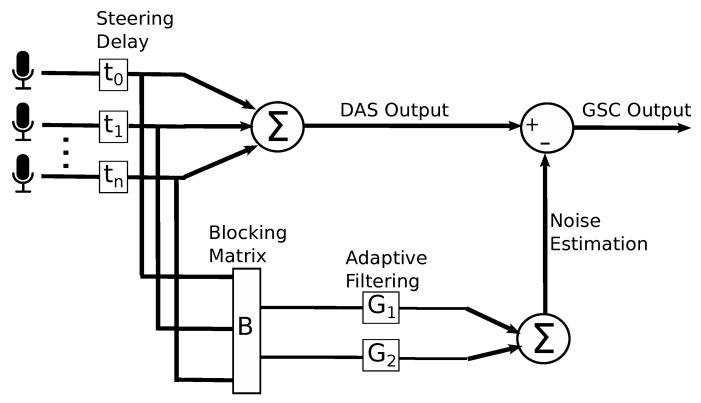
A diagram summarizing Generalized Sidelobe Canceller (GSC).

**Figure 3 sensors-21-05005-f003:**
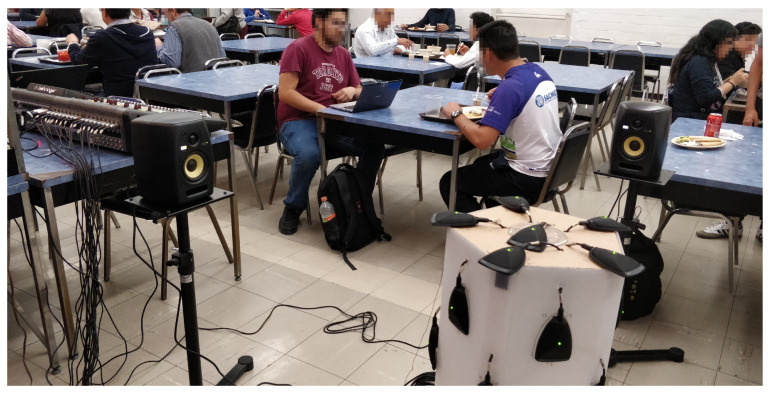
An example of the arrangement of the capture equipment and the position of the sound sources measurements in a real-life setting (Cafeteria).

**Figure 4 sensors-21-05005-f004:**
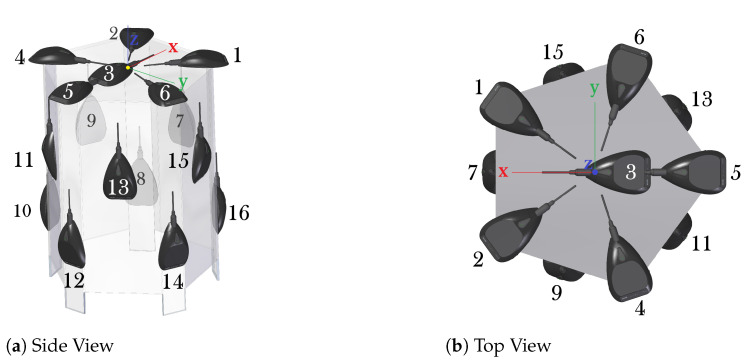
Three-dimensional microphone array of AIRA recordings [6].

**Figure 5 sensors-21-05005-f005:**
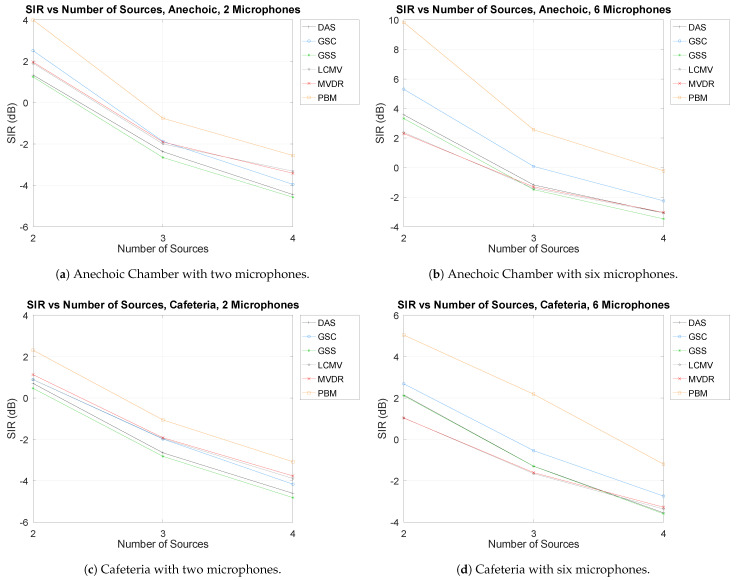
SIR vs. Number of sources.

**Figure 6 sensors-21-05005-f006:**
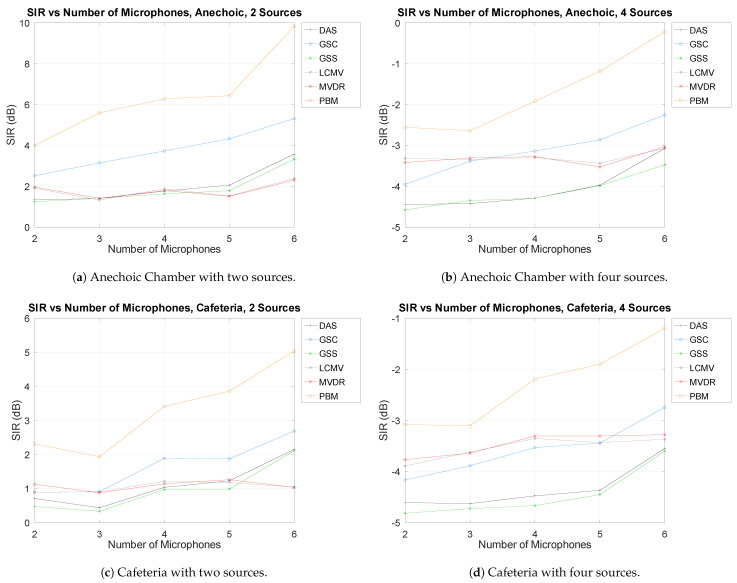
SIR vs. Number of microphones.

**Figure 7 sensors-21-05005-f007:**
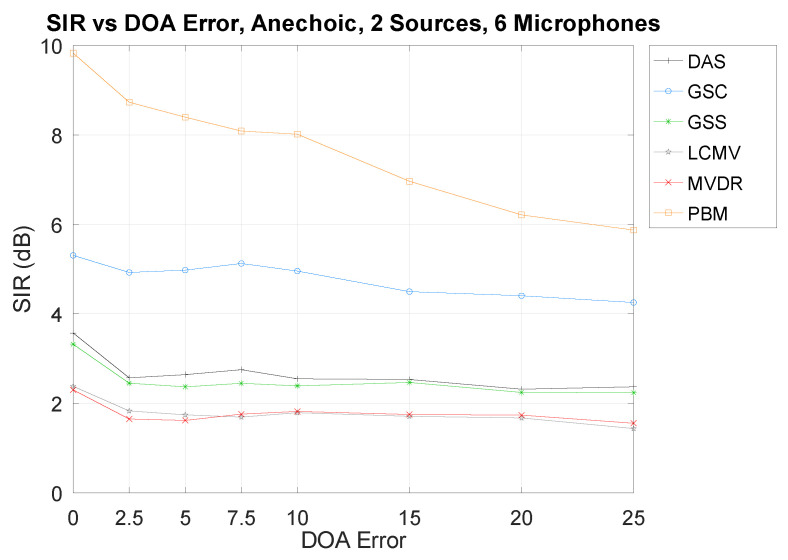
SIR vs. DOA error in the Anechoic Chamber.

**Figure 8 sensors-21-05005-f008:**
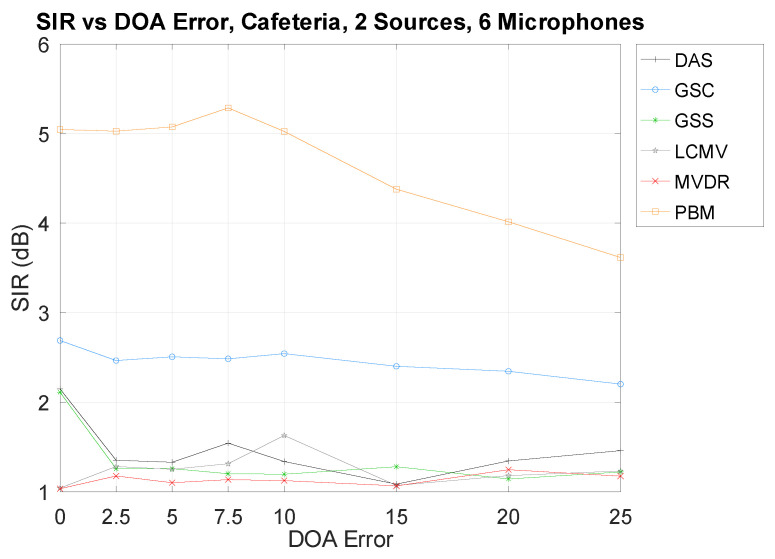
SIR vs. DOA error in the Cafeteria.

**Figure 9 sensors-21-05005-f009:**
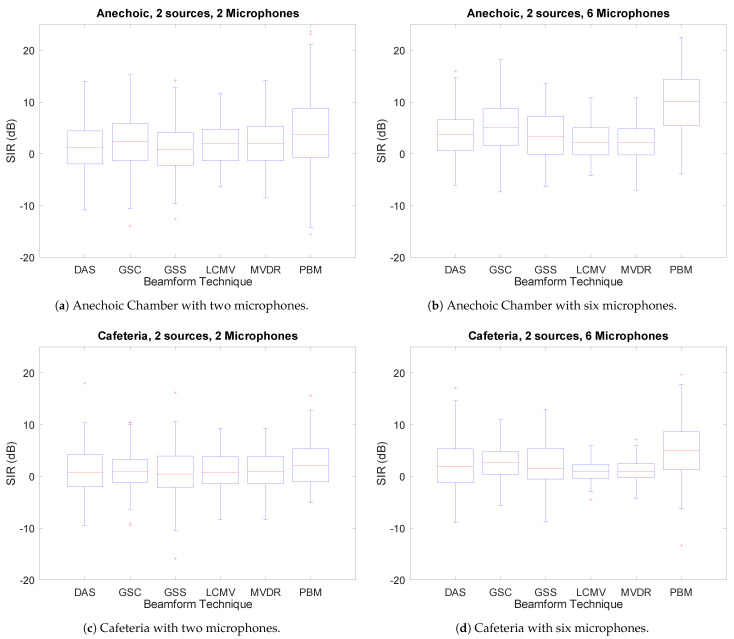
SIR variance for each beamforming technique.

**Table 1 sensors-21-05005-t001:** Microphone positions in the 3D array configuration.

Mic.	x	y
1	0.158	0.115
2	0.158	−0.115
3	−0.045	0.000
4	−0.050	−0.188
5	−0.195	0.000
6	−0.057	0.186

**Table 2 sensors-21-05005-t002:** Overruns triggered; average and standard deviation.

Environment		DAS	GSC	GSS	LCMV	MVDR	PBM
Anechoic	Avg.	0.054	0.035	0.043	5.516	4.469	0.016
	Std.	0.432	0.215	0.289	12.642	10.954	0.145
Cafeteria	Avg.	0.021	0.027	0.044	113.077	106.160	0.020
	Std.	0.245	0.205	0.248	169.970	167.322	0.163

## Data Availability

The dataset used for this study was the Acoustic Interactions for Robot Audition (AIRA), which can be found in https://aira.iimas.unam.mx/ (accessed on 22 June 2021). The implementation of all techniques are open-sourced and available at https://github.com/balkce/beamform (accessed on 22 June 2021).

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
