# Peer review of "A Corpus-Based Evaluation of Beamforming Techniques and Phase-Based Frequency Masking"

_sensors, 2021, doi:10.3390/s21155005_

Round 1
Reviewer 1 Report
There are missing some references for equations 1,2,3..
The sensitivity of used microphone is not mentioned.
What is the purpose of active sources in the experiment?
It is not explained anything about source localization, the distance between sources and from microphone array in different environments?
The directivity pattern of array at different frequencies and different configurations is not shown?
The purpose of the work should be explained better, where this microphone array will be located?
Reviewer 2 Report
In the submitted manuscript, the authors compare performance of different popular beamforming techniques (Delay-and-Sum, Minimum Variance Distortion-less Response, Linearly-Constrained Minimum Variance, Generalized Sidelobe Canceller, Geometric Source Separation, amd Phase-based Binary Masking), under different acoustics contexts (anechoic chamber, cafeteria).
The authors find out (not surprisingly) that the performance of the tested techniques increases with the number of used microphones and that it decreases with the number of interfering sources. As it could be expected, the performance of all the techniques is poorer in the real-life conditions (the cafeteria) compared to the anechoic chamber.
What is surprising is the fact, as the authors report, that the Phase-based Binary Masking technique, despite its simplicity, outperforms the other tested approaches.
The article is clear and well written, the used methodology is described in such a detail that it enables easy replication and further extension. All this makes the submitted manuscript worth being published in Sensors as I believe it finds its readers. However, prior to its acceptance, the following minor issues could be addressed.
- Abstract: The sentence “A systemic evaluation and comparison between different beamforming techniques can benefit the auditory scene analysis community.” Should be reworded as “The auditory scene analysis community can benefit from a systemic evaluation and comparison between different beamforming techniques.”
- Section 3.5: The number of overruns surely depends on the used hardware. Please, provide some related information.
- Line 347: … observed than increasing… -> that
- Line 366: … that the because… (?)
Reviewer 3 Report
the technique we are talking about is widely used in the acoustics sector. It is a mature technique.
As a suggestion I would reduce the initial part of the paper and broaden the discussion.
Highlight what the author's contribution is in this paper.
Explain better how the acoustic measurements were carried out in which rooms, with a description of the dimensions and acoustic characteristics of the rooms.
Perhaps some photos would be appropriate for a better understanding of the activities carried out.
An image of the equipment and the relative arrangement of the measurement set-up.
Type of instrumentation in use
Improve the graphics to make them more readable.
Describe the type of microphones and how they have been calibrated and phased with each other.
The technique of beam forming with two microphones may be insufficient. What is the minimum number of microphones?
An acoustic map of the measured quantities could make the reader better understand the purposes of the paper.
